# Prevalence and risk factors of underweight among under-5 children in Bangladesh: Evidence from a countrywide cross-sectional study

**Md. Moyazzem Hossain** [1,2]*, **Faruq Abdulla** [3]*, **Azizur Rahman** [4]

1 Department of Statistics, Jahangirnagar University, Savar, Dhaka, Bangladesh, 2 School of Mathematics, Statistics and Physics, Newcastle University, Newcastle upon Tyne, United Kingdom, 3 Department of Applied Health and Nutrition, RTM Al-Kabir Technical University, Sylhet, Bangladesh, 4 School of Computing, Mathematics and Engineering, Charles Sturt University, Wagga Wagga, NSW, Australia

* hossainmm@juniv.edu (MMH); faruqiustat09mnil@gmail.com (FA)

## Abstract

### Background

Underweight is one of the important anthropometric components of malnutrition among under-five children and is a major public health concern in Bangladesh because it contributes to mortality as well as morbidity. In designing suitable health programs and policies with the goal of attaining UN SDG Goals, it is necessary to identify the critical risk factors of under-five malnutrition. It is evident that the quantile regression produces robust estimates in the presence of outliers. However, in the context of Bangladesh, no study has been conducted considering the sequential quantile regression on this topic. Therefore, the authors aimed to find the determinants of underweight among under-5 children in Bangladesh considering the latest country representative dataset.

### Methods and materials

The paper considers a weighted sample of size 7762 children are used and data were extracted from the latest Bangladesh Demographic and Health Survey (BDHS) 2017–18 data. Multivariable simultaneous quantile regression models were used to fulfill the objectives of this study.

### Results

Findings depict that undernutrition affects the majority of children in the population as compared to the reference population. The WAZ-score of the child increases by 0.202 points at the 10th quantile of the conditional distribution, and by 0.565 points at the 90th quantile as we move from children of underweight to overweight women. Moreover, the WAZ scores of children from the richest families in the 10th, 50th, and 75th quantiles, respectively, are increased by 0.171, 0.016, and 0.084 points.

**Data Availability Statement:** This study is based on the secondary dataset. One can access the data set via the following link http://dhsprogram.com/data/available-datasets.cfm.

**Funding:** The author(s) received no specific funding for this work.

**Competing interests:** The authors have declared that no competing interests exist.

## Conclusion

Quantile regression revealed the results of several socioeconomic and demographic factors acting differently across the WAZ distribution. Therefore, policymakers may consider the identified risk factors to lessen malnutrition among under-5 children in Bangladesh.

## Introduction

Malnutrition among under-five children is a major public health concern because it contributes to mortality as well as morbidity, particularly in developing countries like Bangladesh where the rate of malnutrition is relatively high. Underweight is one of the indicators of malnutrition. Underweight indicates a nutritional deficiency that has long-term effects on health and consequences on the overall population's well-being. Underweight is determined by the World Health Organization (WHO) Growth Standard weight-for-age (WAZ), less than minus two standard deviations (SD). Between 1990 and 2019, the global prevalence of underweight children under-5 years declined from 24.8 percent to 13 percent. South Asia has a higher prevalence of underweight children than other parts of the world; for example, in 2019, 13.8 percent of children in South Asia were underweight [1]. The latest Multiple Indicator Cluster Survey (MICS) 2019 highlighted that the prevalence of moderate and severe underweight has decreased from 31.9% in 2012–13 to 22.6 percent in 2019 in Bangladesh [2]. Generally, underweight people have been exposed as a risk factor for several diseases such as anemia, hypotension, osteoporosis, osteoporotic fractures in later life, low bone mineral density, reduced sex hormones, feelings of fatigue, and malaise [3, 4]. Being underweight has the potential to have adverse health implications, such as a larger burden of disease, and it has an impact on how many medical disorders will turn out [5–7]. Children's underweight is a serious health condition with substantial consequences for development, health, and well-being [8]. Underweight and related malnourishment have been shown to be especially hazardous for young females [9] and underweight increases the risk of morbidity as well as mortality [10]. Flegal et al. (2018) have assessed that underweight is associated with raised mortality risk in contrast to normal weight in USA population [11]. Do, et al., (2019) noticed that in the Korean population, being underweight was independently linked to a decline in pulmonary function [12]. Park, et al., (2017) show that underweight population had a 19.7% greater risk of cardiovascular disease (CVD) than the normal-weight [13]. Solis-Soto, et al., (2020) observed that children with incomplete immunisation schedules had a considerably greater prevalence of underweight [14].

Shine and Asegidew (2019) show that the prevalence of underweight among children 6–59 months' years old is 48.7% and the sex of the child, the mother's antenatal care non-attendance, the birth interval of infants younger than 24 months, and breastfeeding for infants younger than 12 months are identified to be significant determinants of underweight [15]. A study emphasized the significance of EBF for newborns up to six months old, as it protects them from diarrhoea and ARI-specific morbidity and mortality [16] and more than 35% of the children aged under 5 years suffered from ARI which negatively impacted their health status [17]. Ochiai, et al. (2017) showed that exercise and eating slowly were linked to underweight among Japanese adolescents [18]. Moreover, several previous studies showed that the child's sex, age, size of child at birth, parental educational status, mother's BMI, family income level, mother's antenatal care visit, household water supply, length of nursing, children's birth order, and status of colostrum feeding were linked to underweight [19–38]. Previous research has

primarily examined the correlates of underweight using binary or linear regression approaches. However, the mean effect may overestimate or underestimate the contribution of the covariates at different points and in the presence of outliers [39]. Outliers impacted the estimates of the mean and variance of a dataset [40, 41]. Other epidemiological and health modeling tools and/or medical statistical approaches may overcome these issues. More specifically, the quantile regression (QR) modeling approach yielded more unbiased estimates for skewed data and in the presence of outliers than other regression models [42]. Moreover, several researchers applied the QR model to examine the core socio-demographic factors of a child's nutritional status [43–47].

In the context of Bangladesh, no study has been conducted considering the sequential quantile regression for finding the correlates of underweight among under-5 children in Bangladesh based on the latest country representative dataset. Moreover, employing relevant and cutting-edge statistical approaches to identify key risk factors for under-five malnutrition can assist in developing appropriate health policies and programmes to meet UN SDG Goal 2 target 2. It is apparent that the study data contain outliers, and the sequential quantile regression approach will provide more robust estimates of risk factors for undernutrition than the ordinary least squares regression estimates in presence of outliers because it estimates conditional median instead of conditional mean. It also helps to estimate different percentile of the target variable which provides a clear picture of variability. In order to uncover important risk factors for chronic malnutrition in children under the age of five, this study aims to construct a simultaneous quantile regression, a detailed health and medical statistical model (i.e., underweight) in Bangladesh using the most recent BDHS-2017/18 data.

## Materials and methods

### Data and variables

In this study, the secondary data is obtained from a nationally representative survey called the "2017–18 Bangladesh Demographic and Health Survey (BDHS-2017/18)". The BDHS-2017-18 is the complete survey that covers the enumeration areas (EAs) of the entire country. This survey used stratified sampling and selection is made in two stages. Firstly, 675 EAs were chosen with probability proportional to the size of the EA. In the second phase of selection, 30 households per cluster were carefully chosen with a systematic procedure from the list of households. However, due to natural disaster, data were not collected from the three EAs. These three clusters were in Rajshahi (one rural cluster), Rangpur (one rural cluster), and Dhaka (one urban cluster). The full data set is accessible via the following link http://dhsprogram.com/data/available-datasets.cfm. Before starting the analysis, the authors utilise a weighted sample to ensure that the sample is representative of the country. The details of the sampling procedure and methods of the weighted sample (mathematically adjusted) are available in the published report of BDHS-2017/18 in detail [48]. The weight-for-age Z-score (WAZ) is the target variable, and several child characteristics such as sex (male, female), age (≤6 months, 7–12 months, 13–23 months, 24–35 months, 36–47 months, 48–59 months), duration of breastfeeding (never breastfed, ≤12 months, 13 or more months), birth order (1st, 2nd -3rd, 4th or higher); maternal attributes such as age (≤18 years, 19–24 years, 25–34 years, ≥35 years), educational qualification (no education, primary, secondary or higher) and BMI (underweight (<18.5), normal (18.5–24.9), overweight (≥25)); father's education (no education, primary, secondary or higher), and attributes related to child's health are the explanatory variables in this study. The variables included in this study were chosen based on their availability in the dataset, self-efficacy, as well as related existing research [14, 15, 19–38, 43–47].

**Ethical approval.** Not required because this study considered secondary data which is publicly available. Moreover, the survey was approved by the Ethics Committee of the ICF Macro at Calverton in the USA and by the Ethics Committee in Bangladesh.

## Quantile regression

The quantile regression (QR) model was initially introduced by Koenker and Basset in 1978, and nowadays it is extensively applied in various research areas, particularly in Statistics and Econometrics [49]. Nowadays, it is used in public health [43–47, 49–54]. However, the uses of this technique in public health and epidemiology can provide more reliable statistics and help decisionmakers. Suppose, $Y$ be a random (response) variable having cumulative distribution function (CDF) $F_Y(y)$, i.e. $F_Y(y) = P(Y \leq y)$ and $X$ is the p-dimensional vector of predictor variables. Then the $\tau$th (quantile level) conditional quantile of $Y$ is described as $Q_\tau(Y|X = x) = \{y: F_\tau(y|x)\}$, where $\tau$ varies from 0 to 1.

The QR model portrayed by the conditional $\tau$ th quantiles of the outcome $Y$ for considering the values of predictors $x_1, x_2, \ldots, x_P$ can be expressed as

$$Q_y\left(\tau|x_1, x_2, \ldots, x_p\right) = \beta_0^{(\tau)} + \beta_1^{(\tau)}x_1 + \ldots + \beta_p^{(\tau)}x_p, 0 < \tau < 1, \text{ where}$$

$\beta^{(\tau)} = \left(\beta_0^{(\tau)}, \beta_1^{(\tau)}, \ldots, \beta_k^{(\tau)}\right)^T$ is the vector of unknown parameters.

For a random sample $\{y_1, y_2, \ldots, Y_n\}$ of $Y$, it is understood that the sample median minimises the following sum of absolute deviations, $Median = \arg\min_{\xi \in \mathbb{R}} \sum_{i=1}^{n} \rho_\tau(y_t - \xi)$. Likewise, the general $\tau$ th sample quantile $\xi(\tau)$, that is the equivalent of $Q(\tau)$, is formulated as the minimiser: $\xi(\tau) = \arg\min_{\xi \in \mathbb{R}} \sum_{i=1}^{n} \rho_\tau(y_t - \xi)$, where $\rho_\tau(Z) = Z(\tau - I(Z < 0))$ for $0 < \tau < 1$ denotes the loss function with an indicator function $I(.)$. The loss function $\rho_\tau$ allocates a weight of $\tau$ and 1-$\tau$ for positive residuals $= y_i - \xi$ and negative residuals respectively. The linear conditional quantile function along with this loss function expands the $\tau$ th sample quantile $\xi(\tau)$ to the regression setting in the similar way that the linear conditional mean function expands the sample mean. The OLS estimates is obtained based on the linear conditional mean function $E(Y|X = x) = x'\beta$, by solving $\hat{\beta} = \arg\min_{\beta \in \mathbb{R}^p} \sum_{i=1}^{n} (y_t - x'\beta_i)^2$ [55].

The estimated parameter minimises the sum of squared residuals as the sample mean minimises the sum of squares $\mu = \arg\min_{\mu \in \mathbb{R}} \sum_{i=1}^{n} (y_t - \mu)^2$. Quantile regression also estimates the linear conditional quantile function, $(\tau|X = x) = x'\beta(\tau)$, by solving $\hat{\beta}(\tau) = \arg\min_{\beta \in \mathbb{R}^p} \sum_{i=1}^{n} \rho_\tau(y_t - x'\beta_i)^2$. For any quantile $\tau \in (0,1)$ the quantity $\hat{\beta}(\tau)$ is known as the $\tau$ th regression quantile. For example, $\tau = 0.5$, which minimises the sum of absolute residuals, and also corresponds to $L_1$-type or median regression. The set of regression quantiles $\{\beta(\tau): \in (0,1)$ is called the quantile process [55].

The QR model aimed at solving the term $\min_{\beta \in \mathbb{R}^p}\left[\sum_i \tau|e_i| + \sum_i (1 - \tau)|e_i|\right]$, where $e_i = y_i - x_i'\beta$ is the $i$ th value of unknown errors, $\sum_i \tau|e_i|$ gives the asymmetric penalties $\tau|e_i|$ for over prediction and $\sum_i (1 - \tau)|e_i|$ gives the asymmetric penalties $(1-\tau)|e_i|$ for under prediction

[55]. The $\tau$ th quantile regression estimator $\hat{\beta}(\tau)$ is obtained by minimising the following objective function over $\beta_\tau$

$$Q(\beta_\tau) = \sum_{i \in \{i:y_i \geq x_i'\beta\}}^{N} \tau|y_i - x_i'\beta_\tau| + \sum_{i \in \{i:y_i < x_i'\beta\}}^{N} (1 - \tau)|y_i - x_i'\beta_\tau| \text{where}, 0 < \tau < 1, i : y_i \geq x_i'\beta$$

for over prediction, $i : y_i < x_i'\beta$ for under prediction [55].

The authors considered 0.10, 0.25, 0.50, 0.75, and 0.90 quantile levels and the authors motivated to select these quartiles from previous studies [46, 47, 54, 56, 57].

## Results

Underweight children are those whose weight-for-age is less than two standard deviations (-2 SD) from the reference population's median. Also, if the weight-for-age Z-score of a child is below three standard deviations of the norm, s/he is considered as severely underweight. This study considers the Z-score of weight-for-age as the dependent variable and several demographic, and socioeconomic variables are considered as the explanatory variables. First of all, characteristics of our main concerned variable i.e., the Z-scores of weight-for-age is observed and it can be seen it has some positive skewness (Skewness: 0.62, Kurtosis: 1.14) with mean -1.29 and SD 1.11. The negative average indicates that undernutrition affects the majority of children in the population as compared to the reference population, and the distribution of an index has changed downward [Fig 1(B)]. The boxplot is also used to identify the distribution of the Z-score and the presence of outliers. Therefore, this paper used quantile regression because it provides more robust estimates of the parameters than the ordinary least squares regression in presence of outliers.

The Box plot presented in Fig 1(A), reveals that the distribution of the Z-score is skewed in some extent and there are several outliers in the data set. The prevalence of underweight among under-five Bangladeshi children by selected background characteristics is presented in Table 1. Results depict that around 22.89 percent of children are underweight, and 3.52 percent are severely underweight in Bangladesh.

Male children are slightly less underweight than female children, according to the results shown in Table 1, despite the fact that the percentage of male children is only slightly higher. Given that the prevalence of underweight increased with age, it appeared that child malnutrition was positively correlated with age in children. The child's birth order demonstrated a positive relationship with underweight and a negative association with the child's birth size and interval. As the prevalence of underweight of children is reduced with the increase in mother's nutrition and education levels, the incidence of underweight is negatively correlated with the mother's nutritional status as evaluated by body mass index (BMI) and educational status. It has been found that children who use modern toilets, drink clean water, and have access to electricity are less likely to be underweight than their counterparts. Additionally crucial to a child's nutritional condition is the place of birth. If a child is born at a health facility, there is a decreased likelihood of malnutrition than if the child is born at home. A kid's nutritional status is also correlated with their current state of health since a youngster who has diarrhoea and fever is more likely to be malnourished than a child who is healthy. According to the findings, one-third of children resided in rural areas, where undernutrition in children is more common. In Bangladesh, household amenities are crucial in determining children's nutritional health. In Bangladesh, the relationship between the wealth index and children's nutritional status is strongly positive. We now carry out quantile regression for various quantiles (0.1, 0.25, 0.5, 0.75, and 0.90).

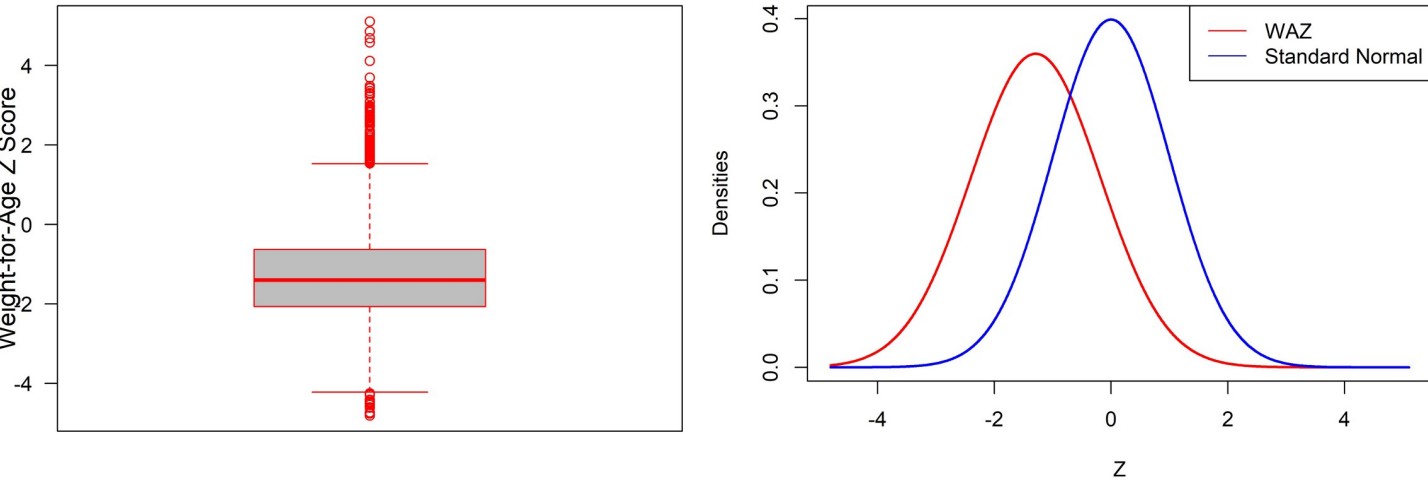

**Fig 1.** Box plot (a) and normal density (b) of weight-for-age Z-score of children.

Table 2 shows that the factors that determine a child's weight for age z-score in Bangladesh are similar. Based on the Z-score of weight-for-age, the sex, and size of the infant at birth are also significant factors in determining their health status. The mother's age, BMI, and level of education greatly influence the child's weight-for-age Z-score at the bottom tail of the conditional Z-score distribution. For instance, the Z-score for weight-for-age increases by 0.246 and 0.043 points at the 90th quantile of the conditional distribution, respectively, when we move from uneducated women to primary, secondary, and above education levels. Additionally, as we move from children of underweight women to those of women with normal BMI, the WAZ-score of the child increases by 0.202 points at the 10th quantile of the conditional distribution, and by 0.565 points at the 90th quantile as we move from children of underweight to overweight women. The conditional distribution of the weight-for-age Z (WAZ) score's upper tail is significantly influenced by household environmental amenities such as clean drinking water, access to power, and bathroom facilities. Compared to children from the poorest families, children from middle-class, wealthy, and wealthy families have higher WAZ scores. For example, the WAZ scores of children from the richest families in the 10th, 50th, and 75th quantiles, respectively, are increased by 0.171, 0.016, and 0.084 points.

## Discussion

Quantile regression analysis was used in this study to identify the crucial risk factors that can help formulate health policies and interventions that will improve child nutritional outcomes and survival. Health is negatively impacted by being underweight; for instance, underweight is associated with a higher mortality risk than people of normal weight [11, 58, 59]. The negative value of WAZ indicates that compared to the reference population, the distribution of an index has changed downward, and the majority of the population's children are undernourished which is consistent with others [50]. Female children are slightly more underweight than their counterparts. Underweight has been shown to be especially hazardous for young females [9]. This might be explained by the fact that in a country like Bangladesh, sons get preference over daughters because they are thought to carry the name and family title of the father and will ensure the future of their parents [60, 61]. A child's age also plays a key role to determine the status of underweight. The proportion of underweight children aged 2–3 years old is more than others. One of the main reason working behind this is that most of the children in

**Table 1. Prevalence of child malnutrition according to underweight by background characteristics.**

| Background characteristics | | Percent | Weight-for-Age (underweight) in % | | p-value of Chi-square |
|---|---|---|---|---|---|
| | | | Z-score <-3 SD | Z-score <-2 SD | |
| Child's sex | Male | 52.16 | 3.25 | 25.42 | 0.102 |
| | Female | 47.84 | 3.80 | 27.48 | |
| Age of the child | ≤ 6 months | 13.14 | 0.10 | 3.64 | <0.001 |
| | 7 months-1 year | 9.94 | 3.58 | 20.64 | |
| | 1–2 years | 18.45 | 5.01 | 31.53 | |
| | 2–3 years | 19.87 | 5.85 | 32.51 | |
| | 3–4 years | 19.16 | 2.57 | 29.59 | |
| | 4–5 years | 19.44 | 2.97 | 30.85 | |
| Birth order | 1st | 38.31 | 3.35 | 24.31 | <0.001 |
| | 2nd-3rd | 49.22 | 3.21 | 25.84 | |
| | 4th or higher | 12.47 | 5.22 | 35.11 | |
| Duration of breastfeeding | Never breastfeed | 41.28 | 2.78 | 30.00 | <0.001 |
| | < = 12 months | 2.05 | 4.82 | 19.88 | |
| | 13 or more | 6.97 | 3.05 | 28.95 | |
| | Still breastfeeding | 49.70 | 4.14 | 23.47 | |
| Mother's age | Up to 18 years | 7.23 | 6.38 | 26.38 | <0.001 |
| | 19–24 | 40.24 | 3.21 | 25.04 | |
| | 25–34 | 44.67 | 3.32 | 26.96 | |
| | 35 or more | 7.86 | 3.70 | 30.59 | |
| Mother's BMI | Underweight (<18.5) | 13.61 | 7.88 | 39.31 | <0.001 |
| | Normal (18.5–24.9) | 59.21 | 3.18 | 26.43 | |
| | Overweight (> = 25) | 27.18 | 1.97 | 19.39 | |
| Mother's Education level | No Education | 7.15 | 6.02 | 39.65 | <0.001 |
| | Primary | 28.40 | 4.82 | 32.64 | |
| | Secondary or above | 64.45 | 2.67 | 22.16 | |
| Father's Education level | No Education | 14.85 | 5.71 | 38.01 | <0.001 |
| | Primary | 34.29 | 4.25 | 30.67 | |
| | Secondary or above | 50.86 | 2.42 | 20.07 | |
| Type of place of residence | Rural | 73.04 | 3.54 | 27.76 | <0.001 |
| | Urban | 26.96 | 3.50 | 22.69 | |
| Religion | Muslim | 91.96 | 3.48 | 26.60 | <0.001 |
| | Non-Muslim | 80.04 | 3.85 | 24.19 | |
| Place of delivery | With Health Facility | 49.91 | 2.97 | 18.99 | <0.001 |
| | Respondent's Home | 50.09 | 4.99 | 29.11 | |
| Number of ANC visits | None | 13.13 | 4.25 | 25.49 | 0.026 |
| | 1–3 | 44.66 | 4.15 | 25.58 | |
| | 4–7 | 36.18 | 4.22 | 22.19 | |
| | 8 or more | 6.03 | 1.45 | 18.55 | |
| Had diarrhea recently | No | 95.26 | 3.49 | 26.27 | 0.043 |
| | Yes | 4.74 | 3.98 | 29.18 | |
| Had fever in last two weeks | No | 66.79 | 3.18 | 24.75 | <0.001 |
| | Yes | 33.21 | 4.18 | 29.66 | |
| Had cough in last two weeks | No | 64.01 | 3.26 | 25.55 | 0.003 |
| | Yes | 35.99 | 3.96 | 27.89 | |

*(Continued)*

**Table 1.** (Continued)

| Background characteristics | | Percent | Weight-for-Age (underweight) in % | | p-value of Chi-square |
|---|---|---|---|---|---|
| | | | Z-score | Z-score | |
| | | | <-3 SD | <-2 SD | |
| Wealth index | Poorest | 21.44 | | | <0.001 |
| | Poorer | 20.33 | 5.13 | 34.79 | |
| | Middle | 18.87 | 4.59 | 32.11 | |
| | Richer | 19.88 | 2.91 | 24.45 | |
| | Richest | 19.48 | 2.66 | 24.21 | |

Bangladesh are breastfed until 24 months and breastfeeding gradually declines with child age [62, 63]. Moreover, as kids become older, their immune systems mature and they learn how to interact with their surroundings more effectively by avoiding germy places and consuming healthy foods, for example [17]. Children who suffered from different childhood morbidities such as fever, diarrhea, and ARI are more underweight than children who do not suffer from them.

Children whose mothers had a secondary or higher education had a lower risk of being underweight than children whose mothers had no education. The results were in line with a study [64]. The reasoning can be that mothers with less education have less experience and understanding with childcare, nutrition, health communication, maintaining a clean environment, nursing, and medical difficulties [17, 65]. Early married women also have fewer opportunities to continue their education, which affects the health of their children [66] and sometimes, wives were physically assaulted by their husbands for neglecting their children [67]. Mother's BMI has an influence on the status of underweight of their children. The WAZ score of a child increases as mother's BMI status shifted from underweight to normal. Moreover, compared to children from wealthy families, those from poorer families were more likely to be underweight. This outcome is consistent with the findings of the WHO indicators study, which found that underweight child prevalence was much lower in families with higher monthly per capita income [19, 68]. Typically, wealthier families can ensure better medical care along with more nutritious food as well as can provide an improved and healthier living environment [37]. Due to inadequate food consumption, a lack of access to basic healthcare, and a higher risk of infection, children from lower-income families were more likely to be underweight than children from higher-income families. Additionally, antenatal care quality is closely related to one's level of wealth. This study supported findings from earlier research conducted in other developing countries, showing that household economic circumstances play a significant role in the nutritional status of children in these countries [63, 69–71].

## Strengths and limitations of this study

This study's novelty and use of the most recent country-representative BDHS-2017/18 dataset are its greatest strengths. This is the first study that used quantile regression in order to measure the determinants of underweight among under-five children in Bangladesh. However, there are some limitations of this study. Firstly, the results from this sample are not transferable to other populations with different characteristics, because socioeconomic factors were only assessed at a one-time point. Secondly, it is crucial to remember that simply because this research is cross-sectional, causal inferences cannot be made. Finally, the prevalence of

**Table 2. Results of quantile regression analysis for Weight-for-Age Z (WAZ) score for under-5 years Bangladeshi children.**

| Characteristics | Labels | Q10 | Q25 | Q50 | Q75 | Q90 |
|---|---|---|---|---|---|---|
| | | Coefficient (95% CI) | Coefficient (95% CI) | Coefficient (95% CI) | Coefficient (95% CI) | Coefficient (95% CI) |
| Child's sex | Male (Ref.) | | | | | |
| | Female | 0.077 (-0.024,0.178) | 0.081* (-0.006,0.168) | 0.025* (-0.003,0.052) | 0.021** (0.006,0.036) | 0.008* (-0.002,0.018) |
| Age of the child | < = 6 months (Ref.) | | | | | |
| | 7 months-1 year | -0.068*** (-0.094,-0.042) | -0.243* (-0.528,0.042) | -0.255* (-0.525,0.015) | -0.062*** (-0.086,-0.037) | -0.772** (-0.183,0.276) |
| | 1–2 years | -0.505*** (-0.797,-0.212) | -0.698*** (-0.994,-0.402) | -0.707*** (-1.091,-0.324) | -0.47*** (-0.719,-0.222) | -0.062* (-0.126,0.001) |
| | 2–3 years | -0.439** (-0.72,-0.158) | -0.704*** (-1,-0.408) | -0.777*** (-1.119,-0.434) | -0.652*** (-0.859,-0.444) | -0.206* (-0.449,0.037) |
| | 3–4 years | 0.317** (0.022,0.612) | 0.168** (0.028,0.307) | 0.141* (-0.018,0.3) | 0.227** (-0.007,0.462) | 0.165** (0.001,0.329) |
| | 4–5 years | -0.195 (-0.539,0.148) | -0.216** (-0.395,-0.036) | -0.208* (-0.435,0.018) | -0.194* (-0.42,0.031) | -0.182** (-0.342,-0.022) |
| Birth order | 1st (Ref.) | | | | | |
| | 2nd-3rd | 0.005 (-0.004,0.014) | -0.015 (-0.092,0.061) | -0.071** (-0.134,-0.008) | -0.117** (-0.202,-0.032) | -0.198** (-0.382,-0.013) |
| | 4th and higher | 0.044* (-0.007,0.096) | 0.003 (-0.121,0.127) | -0.128* (-0.284,0.028) | -0.233** (-0.443,-0.023) | -0.508** (-0.857,-0.158) |
| Duration of breastfeeding | Never breastfeed (Ref.) | | | | | |
| | < = 12 months | -0.023 (-1.183,1.137) | -0.003 (-0.111,0.104) | 0.098* (-0.003,0.199) | -0.204 (-1.509,1.101) | -0.943** (-1.689,-0.197) |
| | 13 or more | -0.112** (-0.213,-0.01) | -0.026* (-0.056,0.004) | -0.175*** (-0.271,-0.08) | -0.438 (-1.673,0.796) | -0.353 (-1.077,0.371) |
| | Still breastfeeding | -0.243** (-0.473,-0.012) | -0.095* (-0.201,0.011) | -0.095** (-0.189,0) | -0.281 (-1.494,0.933) | -0.568* (-1.19,0.055) |
| Religion | Muslim (Ref.) | | | | | |
| | Non-Muslim | -0.006 (-0.13,0.119) | 0.053 (-0.069,0.174) | -0.047 (-0.176,0.082) | -0.081 (-0.264,0.102) | -0.208** (-0.411,-0.006) |
| Mother's age | Up to 18 years (Ref.) | | | | | |
| | 19–24 | 0.192* (-0.009,0.394) | 0.044 (-0.109,0.197) | 0.013* (-0.002,0.028) | 0.036** (0.004,0.068) | 0.124 (-0.06,0.308) |
| | 25–34 | 0.129** (0.028,0.229) | -0.034* (-0.071,0.002) | 0.023** (0.003,0.044) | 0.028* (-0.003,0.06) | 0.139* (-0.03,0.307) |
| | 35 or more | 0.075*** (0.052,0.098) | -0.127* (-0.266,0.012) | -0.049* (-0.103,0.005) | -0.063** (-0.126,0) | 0.993** (0.192,1.794) |
| Mother's Education level | No Education (Ref.) | | | | | |
| | Primary | -0.062** (-0.12,-0.003) | -0.001 (-0.004,0.001) | 0.055* (-0.007,0.117) | 0.101** (0.004,0.198) | 0.246* (-0.04,0.533) |
| | Secondary and above | -0.017* (-0.037,0.003) | 0.048** (0.01,0.086) | 0.119* (-0.013,0.251) | 0.071* (-0.001,0.143) | 0.043** (0,0.086) |
| Mother's BMI | Underweight (<18.5) (Ref.) | | | | | |
| | Normal (18.5–24.9) | 0.202*** (0.098,0.305) | 0.303*** (0.22,0.387) | 0.321*** (0.249,0.393) | 0.326*** (0.2,0.452) | 0.565*** (0.303,0.828) |
| | Overweight (> = 25) | 0.338*** (0.229,0.448) | 0.407*** (0.314,0.499) | 0.403*** (0.308,0.497) | 0.584*** (0.45,0.718) | 0.819*** (0.538,1.1) |
| Father's Education level | No Education (Ref.) | | | | | |
| | Primary | 0.089* (-0.003,0.181) | -0.036** (-0.072,-0.001) | -0.043* (-0.09,0.003) | -0.006 (-0.02,0.008) | 0.025** (0.003,0.048) |
| | Secondary and above | 0.092* (-0.005,0.188) | 0.044** (0,0.088) | 0.002* (0,0.005) | 0.046* (-0.005,0.096) | 0.074* (-0.011,0.158) |

(*Continued*)

**Table 2.** (Continued)

| Characteristics | Labels | Q10 | Q25 | Q50 | Q75 | Q90 |
|---|---|---|---|---|---|---|
| | | Coefficient (95% CI) | Coefficient (95% CI) | Coefficient (95% CI) | Coefficient (95% CI) | Coefficient (95% CI) |
| Type of place of residence | Rural (Ref.) | | | | | |
| | Urban | -0.071 (-0.162,0.021) | 0.015 (-0.093,0.122) | 0.043** (0,0.086) | 0.077* (-0.003,0.157) | -0.069 (-0.291,0.153) |
| Place of delivery | With Health Facility (Ref.) | | | | | |
| | Respondent's Home | -0.001 (-0.113,0.111) | -0.014 (-0.097,0.07) | -0.022 (-0.094,0.05) | -0.076* (-0.16,0.009) | -0.111* (-0.245,0.023) |
| Number of ANC visits | None (Ref.) | | | | | |
| | 1–3 | -0.108** (-0.175,-0.04) | -0.045* (-0.094,0.003) | -0.029 (-0.134,0.077) | -0.098* (-0.215,0.019) | -0.4** (-0.803,0.004) |
| | 4–7 | -0.108** (-0.182,-0.035) | -0.036** (-0.066,-0.005) | -0.038 (-0.161,0.085) | -0.102* (-0.215,0.01) | -0.406* (-0.881,0.069) |
| | 8 or more | -0.082** (-0.153,-0.011) | 0.024** (0.004,0.043) | -0.039 (-0.152,0.074) | -0.034 (-0.321,0.252) | 0.049* (-0.006,0.104) |
| Had diarrhea recently | No (Ref.) | | | | | |
| | Yes | -0.021 (-0.215,0.172) | -0.002 (-0.17,0.167) | -0.014 (-0.136,0.107) | -0.057 (-0.283,0.169) | -0.118 (-0.559,0.323) |
| Had fever in last two weeks | No (Ref.) | | | | | |
| | Yes | -0.219*** (-0.308,-0.13) | -0.215*** (-0.312,-0.119) | -0.21*** (-0.307,-0.113) | -0.208*** (-0.32,-0.095) | -0.103* (-0.224,0.017) |
| Had cough in last two weeks | No (Ref.) | | | | | |
| | Yes | -0.015 (-0.114,0.084) | 0.047** (0.005,0.089) | 0.055** (0.008,0.103) | 0.114** (0.004,0.223) | -0.002 (-0.033,0.028) |
| Received BCG | No (Ref.) | | | | | |
| | Yes | 0.02 (-0.197,0.236) | -0.059 (-0.286,0.169) | -0.157** (-0.31,-0.005) | -0.278** (-0.506,-0.05) | -0.653** (-1.235,-0.07) |
| Received Vitamin A | No (Ref.) | | | | | |
| | Yes | -0.153*** (-0.244,-0.061) | -0.133*** (-0.203,-0.063) | -0.204*** (-0.294,-0.114) | -0.312*** (-0.438,-0.187) | -0.529*** (-0.781,-0.276) |
| Wealth index | Poorest (Ref.) | | | | | |
| | Poorer | 0.152** (0.029,0.274) | 0.071** (0.021,0.12) | -0.066* (-0.136,0.003) | -0.052 (-0.12,0.017) | -0.365** (-0.708,-0.022) |
| | Middle | 0.17** (0.012,0.329) | 0.119** (0.014,0.225) | 0.072* (-0.012,0.156) | -0.011* (-0.024,0.002) | -0.248** (-0.479,-0.018) |
| | Richer | 0.12** (0.005,0.234) | 0.058** (0.005,0.11) | 0.051* (-0.001,0.103) | 0.02** (0,0.041) | -0.129* (-0.269,0.01) |
| | Richest | 0.171** (0.01,0.332) | 0.019* (-0.01,0.048) | 0.016** (0.001,0.032) | 0.084* (-0.01,0.177) | -0.021* (-0.045,0.003) |
| Constant | | -1.486** (-2.619,-0.353) | -0.791 (-1.871,0.29) | 0.097 (-0.994,1.189) | 1.05 (-0.426,2.526) | 1.936 (-0.805,4.677) |

Notes: *** refers p-value <0.001,

** refers p-value <0.05 and

* refers p-value <0.1.

underweight may also vary across time and space, however, the investigators did not take this into consideration in this study.

## Conclusions

Age, and child's size at birth, mother's educational level and BMI, and income quantile all play significant roles in determining a child's WAZ score. It is necessary to focus on children aged less than three years to reduce the burden of underweight in Bangladesh. Improved mothers' education and health status will be helpful to lessen the prevalence of underweight among under-five children because a healthy mother may give birth to healthy children. Moreover, special attention is needed for children who lived in poor families. It seems vital to include information on ways of determining optimal weight in health promotion campaigns because underweight can have a range of harmful implications. Therefore, it would seem necessary to describe the effects of underweight in campaigns for health promotion. The authors would like to suggest that Bangladesh's educational system incorporate advice and guidance concerning nutrition and health education. Moreover, the authors believe that these findings will be helpful to the policymakers to ensure the SDGs goal-3. Furthermore, to observe the trend and get an in-depth scenario temporal and spatial variability should be incorporated into a further study.

## Acknowledgments

The authors are thankful to the authorities of BDHS for providing data. We also grateful to the well-wishers and family members for their inspiration and sacrifice. They also thank the academic editor and two reviewers for their valuable comments and suggestions that helped to enhance the quality of the manuscript.

## Author Contributions

**Conceptualization:** Md. Moyazzem Hossain, Faruq Abdulla, Azizur Rahman.

**Data curation:** Md. Moyazzem Hossain.

**Formal analysis:** Md. Moyazzem Hossain, Faruq Abdulla.

**Methodology:** Md. Moyazzem Hossain, Faruq Abdulla, Azizur Rahman.

**Supervision:** Azizur Rahman.

**Validation:** Azizur Rahman.

**Visualization:** Md. Moyazzem Hossain, Faruq Abdulla.

**Writing – original draft:** Md. Moyazzem Hossain, Faruq Abdulla.

**Writing – review & editing:** Md. Moyazzem Hossain, Faruq Abdulla, Azizur Rahman.

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
