## [Decision Letter · Decision Letter 0]

16 Mar 2023

PONE-D-23-00547Prevalence and risk factors of undernutrition among under-5 children in Bangladesh: evidence from a countrywide cross-sectional studyPLOS ONE

Dear Dr. Hossain,

Thank you for submitting your manuscript to PLOS ONE. After careful consideration, we feel that it has merit but does not fully meet PLOS ONE’s publication criteria as it currently stands. Therefore, we invite you to submit a revised version of the manuscript that addresses the points raised during the review process. Please submit your revised manuscript by Apr 30 2023 11:59PM. If you will need more time than this to complete your revisions, please reply to this message or contact the journal office at plosone@plos.org. Please include the following items when submitting your revised manuscript:A rebuttal letter that responds to each point raised by the academic editor and reviewer(s). You should upload this letter as a separate file labeled 'Response to Reviewers'.A marked-up copy of your manuscript that highlights changes made to the original version. You should upload this as a separate file labeled 'Revised Manuscript with Track Changes'.An unmarked version of your revised paper without tracked changes. You should upload this as a separate file labeled 'Manuscript'.

We look forward to receiving your revised manuscript.

Kind regards,

Jayanta Kumar Bora,PhD

Academic Editor

PLOS ONE

Reviewers' comments:

Reviewer's Responses to Questions

**Comments to the Author**

1. Is the manuscript technically sound, and do the data support the conclusions?

Reviewer #1: Yes

Reviewer #2: Yes

2. Has the statistical analysis been performed appropriately and rigorously? 

Reviewer #1: Yes

Reviewer #2: Yes

3. Have the authors made all data underlying the findings in their manuscript fully available?

Reviewer #1: No

Reviewer #2: Yes

4. Is the manuscript presented in an intelligible fashion and written in standard English?

Reviewer #1: Yes

Reviewer #2: Yes

5. Review Comments to the Author

Reviewer #1: 1. Title of the manuscript is quite confusing. In the title authors mentioned undernutrition but the concept of undernutrition couldn’t be found in the main text. The authors should clearly define the key concepts.

2. The abstract section of the manuscript is good and precise which makes more clear to the readers.

3. Introduction part needs revision. If authors could add more details in the introduction that would be more impactful for the readers. The clarity part seems missing which needs to be added by the revision of the same.

4. In the introduction part it is not clear why the authors only focused on the underweight. What types of undernutrition prevails? What is malnutrition? What is the difference between malnutrition and undernutrition? However, it would be more relevant if the authors could add some more relevant studies which can highlight the gaps in the existing literature. Why and how sequential quantile regression is relevant to minimize the gaps?

5. The detailed data and estimation method section seem fine and authors appreciation for the same.

6. On page number 14, it would be better to start as mothers rather than ‘moms had a secondary or higher education….’. Mom’s word does not sound well.

7. The conclusion section needs refinement. The paper needs to jump swiftly into the conclusion by explaining why all the results are notable and important. However, the authors do not explain the policy implications properly. If authors could do that it would be better.

8. Results and discussion section seems fine and would be appreciated.

9. However, I would like to insist that the authors should explain a bit that what are the study limitation.

Reviewer #2: The topic is very relevant in the context of the increasing undernutrition among children in LMIC, particularly in Bangladesh. However, it needs a few modifications as discussed below.

1. In the abstract, it is necessary to mention the reason for using the quantile regression models.

2. In the Introduction section, it is recommended to update the recent statistics about undernutrition among under-5 children in the global and Bangladesh context.

3. In the Methodology section, the authors mentioned “The variables included in this study were chosen based on their availability in the dataset, self-efficacy, as well as related existing research”. It is strongly recommended to add citations for “existing research”.

4. As per PloS ONE guideline, “Ethical statements” should be placed in the Methods section

5. The Result and Discussion are well written.

6. Add strengths and limitations of this study.

7. The manuscript needs thorough typos and language correction.

8. Check all the references in the reference list carefully and delete unnecessary information for example “[Internet]”, “[cited 2022 Jan 8]”, and so on. Moreover, use “Available from: …” only for online documents.

9- The English should be revised

10- A future work section should be added

6. PLOS authors have the option to publish the peer review history of their article (what does this mean?). If published, this will include your full peer review and any attached files.

Reviewer #1: No

Reviewer #2: No

---

## [Author Response · Author response to Decision Letter 0]

17 Mar 2023

Dear Academic Ediotor,

PLOS ONE

We would like to express our sincere gratitude to the reviewers and the Editors for their valuable comments. We have considered all the comments made by the reviewers and thoroughly revised and formatted the manuscript accordingly. A detailed response to each of the comments is provided below:

Authors Response to the Academic Editor's Comments:

Thank you very much for carefully checking the manuscript and providing insightful comments. 

All required files are uploaded to the journal system. Revised texts are in red color.

Authors Response to the Journal Requirements:

1. Thanks. We revised the manuscript following the PLOS ONE style. Revised texts are in red color.

Page: 1-16

2. Thanks. We add it in the Methods section. Revised texts are in red color.

Page: 5

3. Thanks. We add it in the Methods section. Revised texts are in red color.

Page: 5

Authors Response to the Reviewer 1 Comments:

1. Thank you very much for carefully checking the manuscript and providing insightful comments. 

We revise the title of the manuscript. Revised texts are in red color.

Page: 1

2. Thanks for the inspiration. 

3. We appreciate your comments. We revise the Introduction section. 

Revised texts are in red color. Page: 2-4

4. Thanks. You know, undernutrition includes wasting, stunting, being underweight, and inadequate vitamins or minerals. On the other hand, malnutrition includes stunting, wasting, and underweight. 

We appreciate your concern. Basically, this study focused only on the underweight. Thus, we revise the title of the manuscript.

We also revise the manuscript as per your comments. Revised texts are in red color.

Page: 3-4

5. We are grateful to you for this kind of appreciation. It motivates us to continue research. 

6. We are thankful to you for carefully checking the manuscript. We agreed with you and it is revised. Revised texts are in red color.

Page: 14

7. Thanks for highlighting this point. We revise the Conclusion section. Revised texts are in red color.

Page: 15

8. We are thankful to you for your appreciative comments and feedback. 

9. Thanks. We add the strength and limitations section in the revised version. Revised texts are in red color.

Page: 14-15

Authors Response to the Reviewer 2 Comments:

1. Thanks. We appreciate your comments. We have revised the Abstract section as per your feedback. Revised texts are in red color.

Page: 1

2. Thank you very much. This section is revised as per the comment. Revised texts are in red color.

Page: 2

3. Thanks for pointing out this. We add references as per your suggestion. Revised texts are in red color.

Page: 5

4. Thanks. We move the Ethical statement in the Methods section following the PloS ONE style. Revised texts are in red color.

Page: 5

5. Thank you very much for your appreciative comments. 

6. Thanks. We add the strength and limitations section in the revised version. Revised texts are in red color.

Page: 14-15

7. Thanks. We check and correct all typos and grammatical mistakes. Revised texts are in red color.

Page: 1-15

8. Thank you very much for your careful checking of the manuscript. We use Mendeley for citation and follow the PLOS ONE style for correcting the references. 

9. Thanks. We check and correct all typos and grammatical mistakes. Revised texts are in red color.

Page: 1-15

10. Thank you very much. We add it to the revised manuscript. Revised texts are in red color.

Page: 15

Finally, the revised manuscript has been produced following the valuable comments and suggestions of the reviewers. Once again, we would like to thank the reviewers for their sincere dedication, professional insights, and earnest cooperation in reviewing the manuscript.

---

## [Decision Letter · Decision Letter 1]

10 Apr 2023

Prevalence and risk factors of underweight among under-5 children in Bangladesh: evidence from a countrywide cross-sectional study

PONE-D-23-00547R1

Dear Dr.Moyazzem Hossain,

We’re pleased to inform you that your manuscript has been judged scientifically suitable for publication and will be formally accepted for publication once it meets all outstanding technical requirements.

Kind regards,

Jayanta Kumar Bora,PhD

Academic Editor

PLOS ONE

Additional Editor Comments (optional):

Reviewers' comments:

Reviewer's Responses to Questions

**Comments to the Author**

1. If the authors have adequately addressed your comments raised in a previous round of review and you feel that this manuscript is now acceptable for publication, you may indicate that here to bypass the “Comments to the Author” section, enter your conflict of interest statement in the “Confidential to Editor” section, and submit your "Accept" recommendation.

Reviewer #2: All comments have been addressed

2. Is the manuscript technically sound, and do the data support the conclusions?

Reviewer #2: Yes

3. Has the statistical analysis been performed appropriately and rigorously? 

Reviewer #2: Yes

4. Have the authors made all data underlying the findings in their manuscript fully available?

Reviewer #2: Yes

5. Is the manuscript presented in an intelligible fashion and written in standard English?

Reviewer #2: Yes

6. Review Comments to the Author

Reviewer #2: PLOS ONE

Prevalence and risk factors of underweight among under-5 children in Bangladesh:

evidence from a countrywide cross-sectional study

The paper has been updated

the comments have been addressed

7. PLOS authors have the option to publish the peer review history of their article (what does this mean?). If published, this will include your full peer review and any attached files.

Reviewer #2: No

---

## [Editor Report · Acceptance letter]

12 Apr 2023

PONE-D-23-00547R1 

Prevalence and risk factors of underweight among under-5 children in Bangladesh: evidence from a countrywide cross-sectional study 

Dear Dr. Hossain:

I'm pleased to inform you that your manuscript has been deemed suitable for publication in PLOS ONE. Congratulations! Your manuscript is now with our production department. 

Kind regards, 

on behalf of

Dr. Jayanta Kumar Bora 

Academic Editor

PLOS ONE